# Retrospective study of the immunogenicity and safety of the CoronaVac SARS-CoV-2 vaccine in people with underlying medical conditions

Chunmei Li[1✉], Hanfang Bi[1,16], Zhenwang Fu[2,16], Ao Li[1], Na Wan[1], Jun Hu[3], Fan Yang[1], Tai-Cheng Zhou[4], Yupeng Liang[5], Wei Su[1], Tianpei Shi[1,6,7], Mei Yang[1], Rong Wang[1,6,7], Wanting Qin[1,6,7], Xuanjing Yu[1,6,7], Hong-Yi Zheng[8], Zumi Zhou[1], Yong-Tang Zheng [8], Jia Wei[4], Gang Zeng[5✉], Zijie Zhang [1,4✉] & the Precise-CoVaccine study group*

## Abstract

**Background** People living with chronic disease, particularly seniors (≥60 years old), made up of most severe symptom and death cases among SARS-CoV-2 infected patients. However, they are lagging behind in the national COVID-19 vaccination campaign in China due to the uncertainty of vaccine safety and effectiveness. Safety and immunogenicity data of COVID-19 vaccines in people with underlying medical conditions are needed to address the vaccine hesitation in this population.

**Methods** We included participants (≥40 years old) who received two doses of CoronaVac inactivated vaccines (at a 3–5 week interval) and were healthy or had at least one of 6 common chronic diseases. The incidence of adverse events after vaccination was monitored. Vaccine immunogenicity was studied by determining neutralizing antibodies and SARS-CoV-2-specific T cell responses post vaccination.

**Results** Here we show that chronic diseases are associated with a higher rate of mild fatigue following the first dose of CoronaVac. By day 14–28 post vaccination, the neutralizing antibody level shows no significant difference between disease groups and healthy controls, except for people with coronary artery disease ($p = 0.0287$) and chronic respiratory disease ($p = 0.0416$), who show moderate reductions. Such differences diminish by day 90 and 180. Most people show detectable SARS-CoV-2-specific T cell responses at day 90 and day 180 without significant differences between disease groups and healthy controls.

**Conclusions** Our results highlight the comparable safety, immunogenicity and cellular immunity memory of CoronaVac in seniors and people living with chronic diseases. This data should reduce vaccine hesitancy in this population.

### Plain language summary

People living with chronic diseases, particularly those over the age of 60, are more likely to have severe symptoms and die following SARS-CoV-2 infection. However, many have not been vaccinated during the national COVID-19 vaccination campaign in China due to concerns about vaccine safety and effectiveness. Here we show that the inactivated COVID-19 vaccine, CoronaVac, is as safe in older people with chronic diseases as it is for healthy people. Also, only slightly differences are seen in the immune response of people with diseases compared to healthy people. Overall, our results highlight that the CoronaVac vaccine is safe and effective in people living with chronic diseases.

[1] State Key Laboratory for Conservation and Utilization of Bio-resources and School of Life Sciences, Yunnan University, 650091 Kunming, Yunnan, China. [2] Hainan Center for Disease Control and Prevention, Hainan, China. [3] Outpatient department, The Affiliated Hospital of Yunnan University, 650091 Kunming, Yunnan, China. [4] Central Lab and Liver Disease Research Center, The Affiliated Hospital of Yunnan University, 650091 Kunming, Yunnan, China. [5] Sinovac Biotech Co. Ltd., Beijing, China. [6] State Key Laboratory of Genetic Resources and Evolution, and Yunnan Laboratory of Molecular Biology of Domestic Animals, Kunming Institute of Zoology, Chinese Academy of Sciences, 100049 Beijing, China. [7] College of Life Science, University of Chinese Academy of Sciences, 100049 Beijing, China. [8] Key Laboratory of Animal Models and Human Disease Mechanisms of Chinese Academy of Sciences, KIZ-CUHK Joint Laboratory of Bioresources and Molecular Research in Common Diseases, Kunming Institute of Zoology, Chinese Academy of Sciences, 650223 Kunming, Yunnan, China. [16]These authors contributed equally: Hanfang Bi, Zhenwang Fu. *A list of authors and their affiliations appears at the end of the paper.
✉email: licm89@126.com; zengg@sinovac.com; zijiezhang@ynu.edu.cn

The coronavirus disease 2019 (COVID-19) caused by severe acute respiratory syndrome coronavirus 2 (SARS-CoV-2) continues to result in large number of deaths and remains to be the greatest challenge for public health. As of March 25, 2022, 475 million COVID-19 cases and more than 6 million deaths were confirmed worldwide[1]. Vaccine is considered as a safe and effective way to reduce SARS-CoV-2 infection, COVID-19 mortality and morbidity. Yet, recently emerged omicron variant (B.1.1.529 & BA.2), which exhibited strong immune escaping from immunity elicited by natural infection or vaccination, has been rapidly spreading and replaced the delta as the predominant variant worldwide.

The omicron variant has greater transmissibility but causes less severe disease than other latest variant of concern (VOC). Preliminary data suggest that the omicron causes milder symptom and most patients recovered without specific treatment, particularly for young people[2,3]. This is largely thanks to the immune barrier induced by vaccinations or previous infections[4,5]. Nevertheless, there are over half of seniors aged 60 years or older remain unvaccinated as of March 2022[6]. Death cases analysis in recent wave of outbreak in Hongkong showed that most fetal cases were senior unvaccinated person with chronic diseases[7]. Thus, the large number of unvaccinated seniors are at high risk of severe outcome and may contribute to a non-negligible number of morbidity and mortality if infection surge fast in population-dense region in China. In many countries, senior people with medical conditions are prioritized for vaccination due to extra vulnerability[8–13]. In China, the vaccination among older people, particular those with underlying medical conditions, were lagging behind due to uncertainty of vaccination safety and effectiveness and well-controlled COVID-19 prevalence in the period of the delta prevalence[14–16]. Significant proportion of this vulnerable population had not yet received one or two doses of COVID-19 vaccine. Lessons from the recent wave of the omicron pandemic indicate increasing coverage of vaccination among seniors ≥60 years of age and people with underlying medical conditions are crucial to reduce COVID-19-associated morbidity and mortality, which in turn contribute to the global pandemic prevention and control.

There have been lots of studies showing comparable safety and effectiveness of SARS-CoV-2 vaccine in special population for various types of vaccines (e.g., diabetes, solid organ transplant, autoimmune disease patients)[17–20]. However, systematic evaluation of inactivated SARS-CoV-2 vaccine safety and effectiveness in people with common diseases has been rare. Here, we report a multicenter retrospective clinical study comparing immunogenicity and reactogenicity of the most widely deployed inactivated vaccine, CoronaVac, in 6 common chronic diseases: hypertension, diabetes mellitus (DM), coronary artery disease (CAD), chronic respiratory disease (CRD), obesity, and cancer in comparison with healthy control. For people living with chronic disease especially seniors older than 60 years, the CoronaVac vaccines are as safe as in healthy people. Although the immunogenicity is slightly different in subgroup of some diseases compared with that of the healthy population, the overall trend is consistent. People living with medical condition show comparable SARS-CoV-2-specific T cell response with the healthy control. Our findings highlight the clinical data to address vaccine hesitancy for seniors and people living with chronic diseases.

## Methods

**Study design and participants**. We conducted a multicenter retrospective study in four different study sites (Haikou city, Wenchang city, and Qionghai city, Hainan Province; Kunming city, Yunnan Province; China), aiming to evaluate the immunogenicity and safety of CoronaVac vacancies in people with underlying medical conditions in comparison with age matched healthy control.

We recruited participants who have received 2 doses of CoronaVac inactivated vaccine with 3–5 weeks of dose interval and were at the 14th–28th day after the second dose at the time of enrollment. Participants were eligible if they were 40 years of age or older, healthy, or diagnosed with any of the 6 most common chronic diseases: hypertension, diabetes mellitus (DM), coronary artery disease (CAD), chronic respiratory disease (CRD), obesity, and cancer, and were able to understand and complete the questionnaires. The diagnose of six underlying medical conditions in our study were mainly based on the diagnosis of local hospital according to National clinical practice guidelines in China. Specifically, we screened potential eligible participants through the CDC database of chronic diseases and vaccination. We then recruit the volunteers who registered as healthy or with at least one of the diseases we studied by telephone interview. During the telephone visit, we will further narrow down to people who have a proof of chronic diseases diagnosis of interest from a local hospital. At the first visit, we checked their case history and asked about their treatment and disease status. People were considered as having underlying medical conditions if they were once diagnosed as one of the 5 common chronic diseases (hypertension, DM, CAD, CRD and cancer), and the disease was not in the acute phase during the recruitment, and maintained the regular treatment during vaccination. The obesity was diagnosed as $BMI \geq 28.0\,kg\,m^{-2}$. Furthermore, they were excluded according to established criteria: had been infected by SARS-CoV-2; had received non-CoronaVac vaccine; with severe mental and neurological diseases; with any other factors unsuitable for clinical observation.

Sex matched healthy participants were recruited as the control group. Since underlying disease condition is common in older adults (≥60 years), participants were grouped into adult (40–59 years old) and senior (≥60 years) subgroups to evaluate the effect of age more accurately on the immunogenicity and safety of the inactivated SARS-CoV-2 vaccine. The disease group was further divided into six subgroups based on the six common diseases we concerned.

Written informed consent was obtained from each participant before enrollment. The study protocol and informed consent form were approved by the Committee on Human Subject Research and Ethics of Yunnan University (CHSRE2021021). This study was conducted in accordance with the requirements of Good Clinical Practice of China and the International Conference on Harmonization. This study is registered with ChiCTR.org.cn, ChiCTR2200058281.

**Adverse events collection**. Participants who came for serum tests 14–28 days after two-dose vaccination were assessed for adverse events within vaccination period. Adverse events including solicited and non-solicited. Solicited injection-site adverse events containing pain, induration, swelling, redness, rash and itching; Solicited systemic adverse events including acute allergy reaction, skin & mucosa abnormalities, diarrhoea, anorexia, vomiting, nausea, muscle pain, headache, cough, fatigue, and fever. Participant were first screened for grade 3 (daily activity significantly affected, medical attention required, and hospitalization may be necessary) or grade 4 (potential life threat, daily activity severely limited and hospitalization required) solicited and non-solicited adverse events by researcher. Participants were then requested to fill a survey to report grade 1 (short or mild reactions without interfering daily activity) and grade 2 (daily activity mildly interfered, simple treatment or no treatment was necessary)

solicited and non-solicited adverse events. The reported adverse events were graded according to the China National Medical Products Administration guidelines (https://www.nmpa.gov.cn/xxgk/ggtg/qtggtg/20191231111901460. html).

**Venous blood collection**. Venous blood was collected for neutralizing antibody response assay at day 14–28, 90 and 180 post two-dose vaccination. For all participants, serum samples were collected by Vacutainer SST tubes (BD, USA). For participants at two sites (Haikou and Kunming), we preserved whole blood using Vacutainer EDTA tubes (BD, USA) at room temperature for no longer than 6 h before isolation. Then we separated peripheral blood mononuclear cells (PBMCs) by density-gradient sedimentation using Ficoll (GE) reagent and cryopreserved at −80 °C until testing.

**Assessment of neutralizing antibody responses**. The neutralizing antibodies to live SARS-CoV-2 were quantified at the central lab and liver disease research center of the affiliated hospital of Yunnan university, using the gold standard of antibody titration: 50 μL serum was first inactivated at 56 °C for 30 min and serially diluted with cell culture medium in two-fold steps. The diluted serum was then incubated with equal volume of live wild type SARS-CoV-2 virus (virus titers: 6.0 lgCCID 50 mL$^{-1}$; passage: P5; GenBank number: MT407649.1) for 2 hours at 36.5 °C. Vero cells were then added to the serum-virus mix and incubated at 36.5 °C for 5 days. Neutralizing antibody titer was calculated by the highest dilution number of 50% protective condition[21].

**Assessment of T-cell responses**. PBMCs were tested by activation-induced cell marker (AIM) assay to quantify the T cell responses: SARS-CoV-2 specific mega-pool (MP) consists of commercial peptide pools (GenScript) derived from peptide scan (15 mers with 11 aa overlap) through the entire Spike glycoprotein (Protein ID: P0DTC2), Membrane protein (Protein ID: P0DTC5) and Nucleoprotein (Protein ID: P0DTC9) of the wild type of SARS-CoV-2. Cells were cultured for 24 h in the presence of SARS-CoV-2 specific MP (1 μg mL$^{-1}$) in 96-wells U bottom plates at $1 \times 10^6$ PBMC per well in complete RPMI containing 5% Human AB Serum (Gemini Bioproducts). Negative control was performed by adding an equimolar amount of DMSO, and phytohemagglutinin (Roche, 1 μg mL$^{-1}$) was included as a positive control. For the surface stain, to remove the cell culture medium, cells were washed with staining buffer (BD). Cells were then resuspended and stained with antibody cocktail (Supplementary Table 1) for 30 min at RT in the dark, washed, and resuspended by staining buffer. All sample flow cytometry data was acquired on a Beckman DxFLEX (The gating strategy was showed in Supplementary Fig. 1).

**Outcomes**. The primary safety endpoint was the occurrence of adverse events within 14 days after the first dose and the second dose of the vaccination. The primary immunogenic endpoints were the seropositive rate and the geometric mean titers (GMTs) of neutralizing antibodies to live SARS-CoV-2 virus (wild type) 14–28 days, 90 days, and 180 days after two-dose vaccination. The secondary immunogenic endpoint was cellular responses (as measured by AIM assay) post 90 days and 180 days of two-dose vaccination.

**Statistical analysis**. We assessed the safety endpoints in the safety population, which included all eligible participants. For the immunogenic outcomes, we assessed the primary endpoints in the participants who completed blood collection at day 14–28, 90, and 180 post vaccination, respectively, and has successful

antibody measurements. We assessed the secondary immunogenic endpoints in the participants who completed blood collection at day 90 and 180 post vaccination, respectively, and has successful T cell response measurements. All statistical analyses were performed by R scripts.

For participants enrolled and completed the assays in each group, the age, weight, BMI were described by mean and standard deviation; the gender and nationality were described by the ratio among total participants. For immunogenicity evaluation, we used descriptive statistics (geometrical mean and 95% confidence interval [CI]) to summarize antibody levels, and the GMT of post-immunization neutralizing antibody was analyzed after logarithmic conversion, and the least square mean of GMT of post-immunization neutralizing antibody and the ratio between groups and its 95% confidence interval (95%CI) were calculated for comparison. The positive rate of neutralizing antibody after immunization was calculated for the experimental group and the control group, the bilateral 95%CI was calculated by Clopper-Pearson method, and the difference between groups was statistically tested by the Chi-square test/Fisher exact probability method. At the same time, geometric mean and 95%CI were used to statistically describe the GMT of the immune neutralizing antibodies of participants in each cohort, and logarithmic converted group T test was used to statistically compare the difference between groups. For the T-cell response analysis, raw data were processed and exported by FlowJo_v10.8.0. The detected ratio of SARS-CoV-2-specific CD4$^+$ and CD8$^+$ T cells after immunization was calculated for the experimental group and the control group, the bilateral 95%CI was calculated by Clopper-Pearson method, and the difference between groups was statistically evaluated by the Chi-square test/Fisher exact probability method. The fraction of SARS-CoV-2-specific T cells was described by median and interquartile range (IQR), and the difference between groups was statistically tested by linear regression analysis, considering AIM assay experimental batch effect.

For safety evaluation, in accordance with the protocol, systemic adverse events and local adverse events will be classified and counted. In this study, Treatment Emergent Adverse Events (TEAE) occurred after inoculation were statistically analyzed. Adverse events were expressed by counts and frequency. The number and incidence of all adverse events in each group were calculated, and the differences between groups were statistically compared using Fisher's exact test method. Descriptive statistics were made for the severity, dosage, and occurrence time of adverse events. The adverse events after each dose of inoculation were statistically analyzed. Adverse events for each dose were analyzed based on the safety population of each dose. No serious adverse event has been reported in this study and thus has not been analyzed.

**Reporting Summary**. Further information on research design is available in the Nature Portfolio Reporting Summary linked to this article.

## Results
**Study participants**. Between 5 Jul and 30 Dec 2021, we recruited 1302 participants at the 14–28th day after the second dose of inactivate SARS-CoV-2 vaccination and collected safety surveys. Among them we enrolled 1266 who had completed all survey questions including demographic information, underlying medical conditions (disease type, duration, severity, and control status) as well as occurrence of adverse events after each dose of vaccination. 297 individuals were excluded from this analysis due to receiving two different products of inactivated vaccines. The

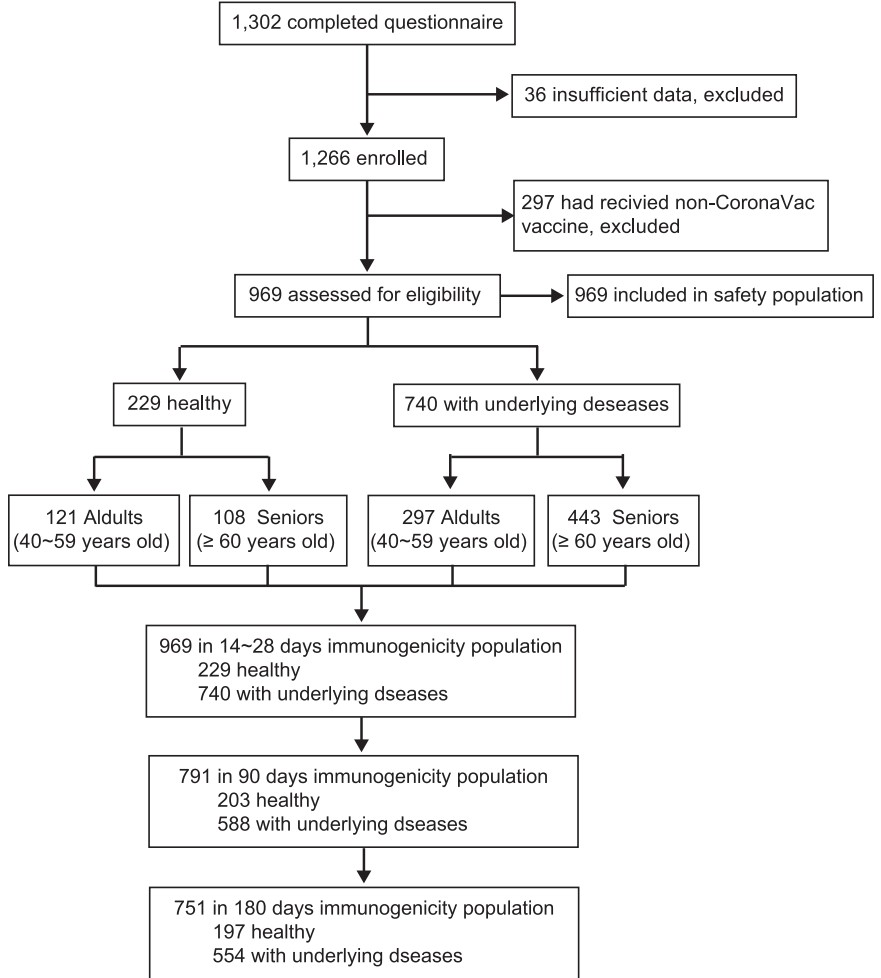

**Fig. 1 Study flow diagram.** Flow chart of recruitment and enrollment (including inclusion/exclusion criteria) for the clinical trial ChiCTR2200058281 to evaluate the immunogenicity and safety of the CoronaVac SARS-CoV-2 vaccine in people with underlying medical conditions.

remaining 969 participants comprised the analyzed samples: 740 people with underlying medical conditions and 229 people as the healthy control. All of them formed the safety population. For the immunogenicity analysis, 969 participants had completed the blood sampling in the 14–28th day after the second dose of vaccine. 178 participants failed to be on site and were excluded from the 90 days immunity assessment, and 40 participants with same reason were excluded from the 180 days analysis (Fig. 1). Each disease group was further divided into 40–59 years old and ≥60 years old subgroups. Baseline demographic characteristics of the comorbidities group and healthy control were well-balanced (Table 1) across age groups, except for the age in ≥60 years old subgroups.

**Vaccine safety**. 150 (20.27%) of 740 comorbidities participants had at least one adverse event, compared with 32 (13.97%) of 229 healthy participants (Supplementary Table 2). Most adverse events were mild (grade 1) in severity and participants recovered within 48 h (Supplementary Table 3). The most frequently reported adverse events in healthy and comorbidities group were injection-site pain (20 [8.73%] of 229 versus 94 [12.70%] of 740), fatigue (9 [3.93%] of 229 versus 48 [6.49%] of 740) and fever (0 [0.00%] of 229 versus 4 [0.54%] of 740). There was no significant difference seen between healthy and comorbidities cohort at overall level (Supplementary Table 2). Although 4 cases of grade 3 adverse events have been reported in 4 individuals in the comorbidities group, including acute allergy, skin & mucosa

abnormalities and fever, most events occurred 7 days after vaccination except for a fever that occurred at the 1st day post vaccination. Thus, none was considered to be related to the vaccination except for the fever, which recovered at the 2nd day post vaccination (Supplementary note 1).

When inspecting the adverse events after the first and the second dose of vaccination, respectively, the incidence of adverse events were 16 (6.99%) of 229 in the health group versus 97 (13.11%) of 740 in the comorbidities group after the first dose, 19 (8.30%) of 229 versus 99 (13.37%) of 740 after the second dose (Supplementary Table 2), with significant difference between health and comorbidities group in both doses ($P = 0.0129$, first dose; $P = 0.0487$, second dose). We then stratified participants by age group: adults (40–59 years old) and seniors (≥60 years old), to explore if seniors exhibit different response to inactivated SARS-CoV-2 vaccine. In the adults subgroup, 81 (27.27%) of 297 participants in the comorbidities group versus 20 (16.53%) of 121 participants in the healthy control, reported adverse events, with statistically significant difference between groups ($P = 0.0231$). Moreover, systemic adverse events of the comorbidities group was significantly higher than that of the healthy control (13.47% versus 4.96%, $P = 0.0099$), while local adverse events showed no significant difference (Table 2). The incidence of adverse events was 54 (18.18%) of 297 in comorbidities people and 9 (7.44%) of 121 in healthy people after the first dose, and 55 (18.52%) of 297 in comorbidities people and 12 (9.92%) of 121 in healthy people after the second dose of vaccine. Both doses showed significant

**Table 1 Demographic and clinical characteristics of the study participants.**

| | Healthy (N = 229) | | Comorbidities (N = 740) | |
| --- | --- | --- | --- | --- |
| | 40-59 years old (N = 121) | ≥60 years old (N = 108) | 40-59 years old (N = 297) | ≥60 years old (N = 443) |
| Age, years | 51.50 (5.55) | 66.89 (5.04) | 53.63 (5.34) | 70.28 (6.54) |
| Height, m | 1.62 (6.98) | 1.61 (9.31) | 1.61 (7.96) | 1.60 (8.53) |
| Weight, kg | 60.20 (9.96) | 61.16 (11.75) | 64.90 (12.62) | 60.95 (10.31) |
| BMI, kg m$^{-2}$ | 22.83 (3.10) | 23.47 (3.42) | 24.83 (3.97) | 23.71 (3.55) |
| Sex, M/F | 55/66 | 42/66 | 152/141 | 187/256 |
| Han nationality | 121 (100.00%) | 108 (100.00%) | 285 (95.96%) | 435 (98.19%) |
| Comorbidities | | | | |
| Coronary artery disease (CAD) | – | – | 31 (10.10%) | 87 (19.86%) |
| Hypertension | – | – | 101 (30.64%) | 131 (31.83%) |
| Diabetes mellitus (DM) | – | – | 71 (22.56%) | 106 (24.83%) |
| Chronic respiratory disease (CRD) | – | – | 31 (9.43%) | 63 (14.90%) |
| Obesity | – | – | 16 (5.39%) | 15 (3.39%) |
| Cancer | – | – | 47 (15.15%) | 41 (9.71%) |

Data are mean (SD) or n (%) unless otherwise specified.

**Table 2 Incidence of adverse events after vaccination in healthy group and comorbidities group.**

| | Healthy (N = 229) | | Comorbidities (N = 740) | | Total (N = 969) | | P value[a] | |
| --- | --- | --- | --- | --- | --- | --- | --- | --- |
| | 40-59 years old (N = 121) | ≥60 years old (N = 108) | 40-59 years old (N = 297) | ≥60 years old (N = 443) | 40-59 years old (N = 418) | ≥60 years old (N = 551) | 40-59 years old | ≥60 years old |
| Total adverse events | 20 (0.17) | 12 (0.11) | 81 (0.27) | 69 (0.16) | 101 (0.24) | 81 (0.15) | 0.0231 | 0.2895 |
| Local reactions | 17 (0.14) | 5 (0.05) | 61 (0.21) | 41 (0.09) | 78 (0.19) | 46 (0.08) | 0.1302 | 0.1722 |
| Pain | 15 (0.12) | 5 (0.05) | 59 (0.20) | 35 (0.08) | 74 (0.18) | 40 (0.07) | 0.0893 | 0.3035 |
| Induration | 0 (0.00) | 0 (0.00) | 5 (0.02) | 3 (0.01) | 5 (0.01) | 3 (0.01) | 0.3275 | 1.0000 |
| Swelling | 3 (0.02) | 0 (0.00) | 5 (0.02) | 5 (0.01) | 8 (0.02) | 5 (0.01) | 0.6956 | 0.5886 |
| Redness | 0 (0.00) | 0 (0.00) | 2 (0.01) | 1 (0.00) | 2 (0.00) | 1 (0.00) | 1.0000 | 1.0000 |
| Rash | 0 (0.00) | 0 (0.00) | 2 (0.01) | 1 (0.00) | 2 (0.00) | 1 (0.00) | 1.0000 | 1.0000 |
| Itching | 0 (0.00) | 0 (0.00) | 3 (0.01) | 6 (0.01) | 3 (0.01) | 6 (0.01) | 0.5599 | 0.6031 |
| Systemic reactions | 6 (0.05) | 7 (0.06) | 40 (0.13) | 33 (0.07) | 46 (0.11) | 40 (0.07) | 0.0099 | 0.8383 |
| Acute allergy | 0 (0.00) | 0 (0.00) | 3 (0.01) | 4 (0.01) | 3 (0.01) | 4 (0.01) | 0.5599 | 1.0000 |
| Skin & mucosa abnormalities | 0 (0.00) | 0 (0.00) | 1 (0.00) | 4 (0.01) | 1 (0.00) | 4 (0.01) | 1.0000 | 1.0000 |
| Diarrhoea | 1 (0.01) | 0 (0.00) | 2 (0.01) | 1 (0.00) | 3 (0.01) | 1 (0.00) | 1.0000 | 1.0000 |
| Anorexia | 0 (0.00) | 1 (0.01) | 4 (0.01) | 1 (0.00) | 4 (0.01) | 2 (0.00) | 0.3284 | 0.3539 |
| Vomiting | 0 (0.00) | 0 (0.00) | 2 (0.01) | 1 (0.00) | 2 (0.00) | 1 (0.00) | 1.0000 | 1.0000 |
| Nausea | 0 (0.00) | 0 (0.00) | 5 (0.02) | 1 (0.00) | 5 (0.01) | 1 (0.00) | 0.3275 | 1.0000 |
| Muscle pain | 1 (0.01) | 0 (0.00) | 12 (0.04) | 6 (0.01) | 13 (0.03) | 6 (0.01) | 0.1203 | 0.6031 |
| Headache | 0 (0.00) | 1 (0.01) | 7 (0.02) | 5 (0.01) | 7 (0.02) | 6 (0.01) | 0.2005 | 1.0000 |
| Cough | 1 (0.01) | 0 (0.00) | 4 (0.01) | 2 (0.00) | 5 (0.01) | 2 (0.00) | 1.0000 | 1.0000 |
| Fatigue | 3 (0.02) | 6 (0.06) | 26 (0.09) | 22 (0.05) | 29 (0.07) | 28 (0.05) | 0.0199 | 0.8075 |
| Fever | 0 (0.00) | 0 (0.00) | 2 (0.01) | 2 (0.00) | 2 (0.00) | 2 (0.00) | 1.0000 | 1.0000 |

[a]The p-value was calculated by Fisher's exact probability method.

differences between diseases and healthy control (P = 0.0062 and P = 0.0387). In the senior subgroup, comorbidities and healthy group did not show significant difference in the overall (15.58% versus 11.11%, P = 0.2895), the first dose (9.71% versus 6.48%, P = 0.3537) or the second dose (9.93% versus 6.48%, P = 0.3543) vaccination. Thus, the higher incidence of adverse events in the adult comorbidities population was the main driving factor for the difference between the comorbidities cohort and healthy control in the overall incidence of adverse events. More specifically, the major inter-group difference was contributed by the systemic adverse events, mainly fatigue (26 [8.75%] of 297 in comorbidities, 3 [2.48%] of 121 in health, P = 0.0199). Furthermore, the incidence of fatigue in adults group was 20 (6.73%) of

297 in the comorbidities group after the first dose, with significant difference from the health control (1 [0.83%] of 121, P = 0.0115); There was no significant difference between comorbidities and health groups after the second dose with 12 (4.04%) of 297 in the comorbidities group and 2 [1.65%] of 121 in the health control (P = 0.3678).

Next, we compared the incidence of adverse events between comorbidities and healthy control stratified by disease types (Supplementary Table 4). The overall incidence of adverse events was 46 (19.83%) of 232 in the hypertension group, 24 (20.34%) of 118 in the CAD group, 34 (19.21%) of 177 in the DM group, 22 (23.40%) of 94 in the CRD group, 19 (21.59%) of 88 in the cancer group, and 5 (6.13%) of 31 in the obesity group versus 32

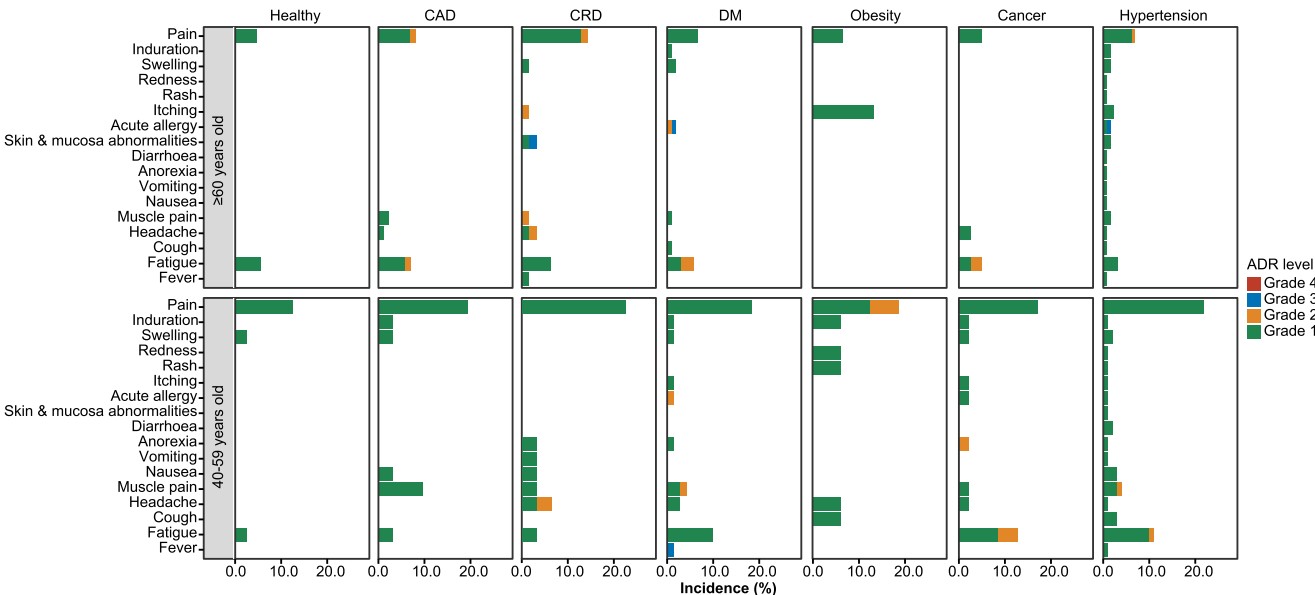

**Fig. 2 Incidence of adverse events reported within 14 days post the first dose and the second dose of the vaccination in the safety population, across age groups.** $n = 969$ study participants. Adverse events post 14 days of first dose and the second dose of the vaccination were collected and graded according to the China National Medical Products Administration guidelines. The histogram shown the incidence of adverse events happened in the 40–60 and ≥60 years old groups. Green, grade 1; orange, grade 2; blue, grade 3; Red, grade 4.

(13.97%) of 229 in healthy control. The most frequently reported adverse events in six disease groups were same as those in the health control, mainly injection-site pain, and fatigue (Fig. 2). All adverse events showed no significant difference between six disease subgroups and health control. When focusing on seniors, the CRD group showed significantly higher incidence of injection-site pain (9 [14.29%] of 63 vs. 5 [4.63%] of 108 in the health control, $P = 0.0404$) than that of the healthy control. Regarding adults, fatigue in the hypertension group (11 [10.89%] of 101 versus 3 [2.48%] of 121 in health group, $P = 0.0124$), DM group (7 [9.86%] of 71, $P = 0.0403$) and cancer group (6 [12.77%] of 47, $P = 0.0152$), and headache in the CRD group (2 [6.45%] of 31 versus 0 [0.00%] of 121 in health group, $P = 0.0405$) showed significant differences compared to the healthy control.

**Humoral immunogenicity.** The immunogenicity of people with underlying diseases and healthy control was evaluated at 14–28 days, 3 months, and 6 months after the two-dose vaccination (Table 3). By day 14–28, the seroconversion rates of neutralizing antibodies were 99 (84%) of 118 in the CAD group, 206 (89%) of 232 in the hypertension group, 150 (85%) of 177 in the DM group, 73 (78%) of 94 in the CRD group, 28 (90%) of 31 in the obesity group, and 75 (85%) of 88 in the cancer group versus 204 (89%) of 229 in health group; The neutralizing antibody GMTs were 22.77 (95% CI 18.29–28.35), 33.89 (95% CI 29.08–39.49), 26.45 (95% CI 21.99–31.82), 22.44 (95% CI 17.33–29.06), 32.92 (95% CI 20.89–51.88), 31.96 (95% CI 24.00–42.56) versus 30.50 (95% CI 26.41–35.22), respectively. Most diseases groups showed no difference in seroconversion rates and GMTs from the healthy control, while CAD ($P = 0.0287$) and CRD group ($P = 0.0416$) exhibited statistically significant decrease of humoral immune response. By day 90, both the seroconversion rates and GMTs were significantly reduced in all people, with no difference between disease and health groups. By day 180, seroconversion rates and GMTs continued to drop, but declined at a much slower rate. Interestingly, cancer patients showed a significant elevation in the seroconversion rate (68% versus 46%, $P = 0.0020$) and GMT (11.50

versus 7.42, $P = 0.0050$) compared to the healthy control (Table 3).

In the adult subgroup (40–59 years old) analyses, GMTs of neutralizing antibodies showed elevations in the cancer (53.71 versus 28.71, $P = 0.0016$) and hypertension (49.02 versus 28.71, $P = 0.0009$) patients compared to healthy control (Supplementary Table 5). Interestingly, in the senior subgroup (≥60 years old), cancer patients (18.56 versus 32.44, $P = 0.0200$) showed the opposite trend with a significantly reduced GMT level. Similarly, senior CAD (20.05, $P = 0.0052$) and CRD (19.85, $P = 0.0144$) patients also showed the same trend of immunogenicity reduction (Supplementary Table 6). Notably, these inter-group variations were no longer significant at 3- and 6-months post vaccination.

Comparing across age subgroups, we observed distinct patterns for disease and health groups. In healthy participants, the seroconversion and GMT of neutralizing antibodies were slightly higher in seniors (≥60 years old) than their younger counterpart (40–59 years old) on day 14–28 and 90 after the second dose. Conversely, senior people with chronic diseases had a lower neutralizing antibody level compared to their younger counterpart by day 14–28 post vaccination. These differences between age groups in seroconversion rate and GMT of neutralizing antibodies diminished by 3 and 6 months after the second dose (Fig. 3a). The reverse cumulative distribution plot showed that, at each time point, the overall distribution of neutralizing antibody titers was generally close across diseases in the senior and adult subgroups (Fig. 3b).

**Cellular immunogenicity.** The SARS-CoV-2-specific T-cell response was quantified utilizing an AIM assay at 3 and 6 months post the two-dose vaccination (Fig. 4). By day 90, the SARS-CoV-2-specific CD4+ T cell responses (OX40+ CD137+) were detected in 21 (100%) of 21 in the CAD group, 31 (86.1%) of 36 in the hypertension group, 21 (95.5%) of 22 in the DM group, 18 (100%) of 18 in the CRD group, 8 (88.9%) of 9 in the cancer group versus 50 (89.3%) of 56 in health control (Supplementary Table 7). The median fraction of SARS-CoV-2-specific CD4+ T cells among CD4+ T cells were 0.05% (IQR 0.019%–0.1%),

**Table 3 Immunogenicity among participants with underling disease and healthy 14–28 days, 3 months, and 6 months after two-dose vaccination.**

|  | Healthy | CAD | Hypertension | DM | CRD | Obesity | Cancer |
|---|---|---|---|---|---|---|---|
| **Day 14–28** |  |  |  |  |  |  |  |
| Seroconversion (%) | 89.00 | 84.00 | 89.00 | 85.00 | 78.00 | 90.00 | 85.00 |
| 95% CI[a] | (0.84, 0.93) | (0.76, 0.90) | (0.84, 0.93) | (0.79, 0.90) | (0.69, 0.86) | (0.74, 0.98) | (0.76, 0.92) |
| P value | - | 0.1800 | 1.0000 | 0.2900 | 0.0200 | 1.0000 | 0.3400 |
| GMT | 30.50 | 22.77 | 33.89 | 26.45 | 22.44 | 32.92 | 31.96 |
| 95% CI[b] | (26.41,35.22) | (18.29,28.35) | (29.08,39.49) | (21.99,31.82) | (17.33,29.06) | (20.89,51.88) | (24.00,42.56) |
| P value | - | 0.0300 | 0.3200 | 0.2300 | 0.0400 | 0.7500 | 0.7700 |
| **Day 90** |  |  |  |  |  |  |  |
| Seroconversion (%) | 55.00 | 52.00 | 58.00 | 52.00 | 51.00 | 65.00 | 61.00 |
| 95% CI[a] | (0.48, 0.62) | (0.41, 0.64) | (0.50, 0.65) | (0.43, 0.60) | (0.39, 0.63) | (0.44, 0.83) | (0.49, 0.73) |
| P value | - | 0.6900 | 0.6100 | 0.5200 | 0.5800 | 0.4000 | 0.4000 |
| GMT | 8.16 | 6.96 | 8.84 | 7.54 | 7.65 | 8.73 | 9.69 |
| 95% CI[b] | (7.23,9.20) | (5.73,8.46) | (7.80,10.00) | (6.51,8.73) | (6.15,9.51) | (6.10,12.49) | (7.52,12.49) |
| P value | - | 0.1700 | 0.3600 | 0.4100 | 0.6100 | 0.7200 | 0.2200 |
| **Day 180** |  |  |  |  |  |  |  |
| Seroconversion (%) | 46.00 | 46.00 | 51.00 | 42.00 | 52.00 | 62.00 | 68.00 |
| 95% CI[a] | (0.39, 0.53) | (0.35, 0.59) | (0.44, 0.59) | (0.34, 0.50) | (0.40, 0.65) | (0.41, 0.80) | (0.55, 0.79) |
| P value | - | 1.0000 | 0.3100 | 0.5100 | 0.3900 | 0.1500 | 0.0020 |
| GMT | 7.42 | 7.39 | 7.73 | 6.60 | 7.37 | 9.16 | 11.50 |
| 95% CI[b] | (6.54,8.42) | (5.91,9.24) | (6.66,8.98) | (5.64,7.73) | (5.85,9.28) | (5.87,14.30) | (8.70,15.19) |
| P value | - | 0.9800 | 0.6800 | 0.2500 | 0.9600 | 0.3600 | 0.0050 |

Seroconversion is considered positive when neutralizing antibody titer to live virus is 1:8 or higher.
[a]Estimated using the Clopper-Pearson method.
[b]Estimated using the Miettinen-Nurminen method.

0.04% (IQR 0.012%–0.084%), 0.08% (IQR 0.02%–0.17%), 0.08% (IQR 0.03%–0.12%), 0.03% (IQR 0.01%–0.12%), and 0.05% (IQR 0.02%–0.11%) in the CAD, hypertension, DM, CRD, cancer, and health group, respectively (Supplementary Table 8). There was no significant difference between disease groups and healthy control on either SARS-CoV-2-specific CD4$^+$ T cell positive rate or fraction. The SARS-CoV-2-specific CD8$^+$ T cell responses (CD69$^+$ CD137$^+$) were detected in 8 (38.1%) of 21 in CAD, 14 (38.9%) of 36 in hypertension, 16 (72.7%) of 22 in DM, 11 (61.1%) of 18 in CRD, 6 (66.7%) of 9 in cancer versus 34 (60.7%) of 56 in the health control (Supplementary Table 7). The median fraction of SARS-CoV-2-specific CD8$^+$ T cell among CD8$^+$ T cell were 0% (IQR 0%–0.01%), 0% (IQR 0%–0.01%), 0.02% (IQR 0%–0.06%), 0.01% (IQR 0%–0.03%), 0.01% (IQR 0%–0.02%), and 0.01% (IQR 0%–0.03%) in the CAD, hypertension, DM, CRD, cancer, and health group, respectively. There were no significant difference between disease groups and healthy control on either SARS-CoV-2-specific CD8$^+$ T cell positive rate or fraction (Supplementary Table 8); By day 180, the detected positive rate and cell fraction of SARS-CoV-2-specific CD4$^+$ and CD8$^+$ T cells were close to that of day 90, with no significant difference between disease groups and healthy control, except for the detected positive rate of the SARS-CoV-2-specific CD4$^+$ T cell in the cancer group at day 180 (81.8% versus 97.6% in health control, $P = 0.0440$), with significant lower than the health control.

## Discussion

People living with underlying medical condition, especially seniors, are at high risk for severe COVID-19-related complications and mortality. Considering the risk vs. benefit, vaccination against SARS-CoV-2 should be prioritized for this special population. Guidelines from the USA, the UK, the Korean and the WHO recommend COVID-19 vaccination for patients with comorbidities. However, safety data for the application of inactivated vaccine, the most widely administered vaccine type in China and many countries, in this vulnerable population has been rare. The lack of data on this special population partially contributed to the vaccine hesitancy among people aged 60 or older in China.

To address vaccine hesitation in seniors and with underlying medical conditions[22,23], we present the first large cohort study addressing the safety and immunogenicity of CoronaVac, the most widely distributed inactivated vaccine, among population living with underlying medical conditions compared with the healthy control. There were no severe adverse events reported in this study. There were one grade 3 fever related to the vaccination, which resolved in 2 days post vaccination. We didn't find significant difference in the incidence of adverse events between the disease population and the healthy control in most of the diseases and age subgroups. We observed elevated incidence of adverse events at certain age subgroup in some diseases, in which the differences were mainly contributed by injection-site pain and fatigue. In addition, these adverse events were mostly mild (grade 1) with 1–2 days resolution time without intervention.

For the immunogenicity, there were no significant differences in most of the disease and age subgroups compared with the healthy control. Although a few diseases showed statistically significant lower neutralizing antibody titers than control in certain age subgroups, the differences were <40% and such differences diminished over time. Thus, the immunogenicity of CoronaVac in people with chronic diseases, particularly in people over 60 years of age, was comparable with healthy counterparts. More importantly, the general trend of slightly lower antibody response in seniors emphasized the importance of this population to receive timely booster doses upon completion of primary immunization.

We observed a significant reduction of neutralizing antibody titers (humoral immunity) 3 months after the second dose. The seroconversion rates of neutralizing antibodies were close to 50% by day 90, and mostly lower than 50% by day 180 post the two-dose vaccination. Moreover, neutralizing antibody levels for some seropositive individuals were too low to provide effective protection against infection[24]. However, in addition to neutralizing antibodies, robust SARS-CoV-2-specific T-cell responses and

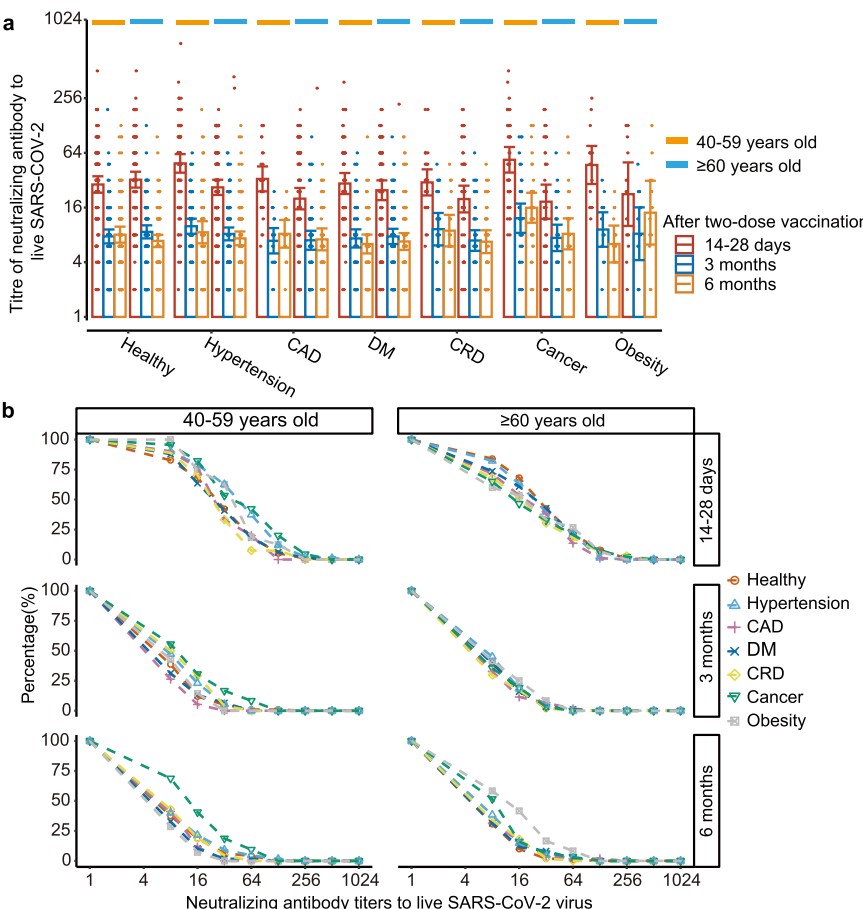

**Fig. 3 Neutralizing antibodies to live SARS-CoV-2 virus (wild type) induced 14–28 days, 90 days, and 180 days after two-dose CoronaVac.**
$n = 969$ study participants. The error bars represent the standard error of the experiment results. **a** Titers of neutralizing antibodies across age groups. red-, blue- and orange-color refer to 14–28 days, 3 months and 6months after two-dose vaccination, respectively. Orange line refer to 40–59 years old group, and blue line refer to senior group older than 60 years old. **b** The inverse cumulative distribution of neutralizing antibodies in hypertension (light blue triangle), CAD (pink plus), DM (blue cross), CRD (yellow rhombus), cancer (green triangle), obesity (gray square) subgroups, and healthy control (red circle), across age groups.

T-cell memory play a critical role in the long-term immune protection against COVID-19, particularly severe symptoms[25]. Thus, we quantified the SARS-CoV-2-specific CD4+ and CD8+ T cell responses by day 90 and 180 utilizing the AIM assay, with a stimulation pool containing peptides covering the entire Spike, M, and N protein. Our data indicates that SARS-CoV-2-specific CD4+ T cells and CD8+ T cells were detectable in most individuals up to 6 months post vaccination and showed no significant difference from the healthy control for most diseases groups. Although the rate of individuals with detectable SARS-CoV-2-specific CD4+ T cells were significantly lower in the cancer group than the health control at day 180, the fraction of SARS-CoV-2-specific CD4+ T cells were still higher than people who had no SARS-CoV-2 vaccination or infection history. Thus, most participants had a substantial CD4+ and CD8+ T-cell responses against SARS-CoV-2 and were maintained at a relative stable level to 6 months after vaccination (the latest time point we assayed) despite of the significant reduction in the neutralizing antibody titers 3 month after the vaccination. This is consistent with the test-negative studies that inactivated SARS-CoV-2 vaccines provided poor protection against symptom over time but good protection against severe symptoms[26].

For the widely used COVID-19 vaccines, some had recruited participants with underlying medical conditions in the phase 3 clinical trials. For example, the Pfizer COVID-19 mRNA vaccine

(BNT162b2) included 20.3% of the sample population reporting one or more underlying diseases. The most common comorbidities were hypertension (24.5%), diabetes mellitus (DM) (7.8%), and chronic lung disease (7.8%). Patients with comorbidities showed similar protection effects and adverse events with the total sample population[27,28]. In the phase 3 clinical trials (P301) for the mRNA-1273 (Moderna), 22.2% of the sample population had high-risk underlying medical conditions. The most common comorbidities were DM (9.4%), severe obesity with body mass index (BMI) $\geq 40\, kg\, m^{-2}$ (6.5%), severe heart disease (4.9%), and CRD (4.8%). The incidence of adverse events was similar in the group with comorbidities and the total sample population with comparable preventive effects[27,29]. Our analyses for CoronaVac, the most widely administered inactivated SARS-CoV-2 vaccine, demonstrated consistent trend with those analyzed in mRNA vaccine trials. Furthermore, the real-world study in Hongkong showed two doses of CoronaVac vaccine had 69.9% [64.4–74.6] severe disease and death protection effectiveness among adults aged 60 years or older, and three doses of CoronaVac had provided high levels of protection against severe or fatal outcomes (97.9% [97.3–98.4])[30]. For people with underlying medical conditions, the real-world studies for patient with DM and chronic kidney diseases all showed no increased risk of adverse events and showed high vaccine effectiveness (VE) against severe and mortality of COVID-19 disease post two-dose CoronaVac

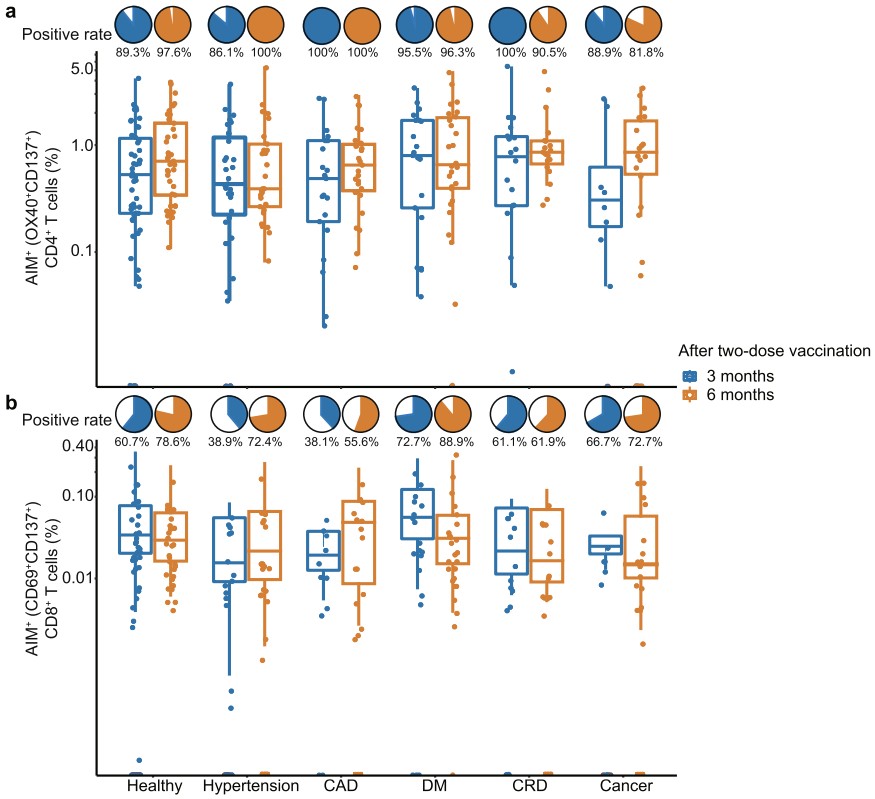

**Fig. 4 The SARS-CoV-2-spesific T cell responses after 3 months, and 6 months after two-dose CoronaVac.** $n = 229$ study participants. Box indicates 25th, 50th, and 75th percentile absolute error, and whiskers indicate 5th and 95th percentile absolute error. Blue color refer to 3 months after two-dose vaccination, while orange color refer to 6 months. The detected positive rate and cell fraction of SARS-CoV-2-specific CD4+ T cell responses (OX40+ CD137+) (**a**) and the SARS-CoV-2-specific CD8+ T cell responses (CD69+ CD137+) (**b**) in different chronic diseases subgroups and healthy control, quantified by AIM assays.

vaccine[31,32]. The results of these real-world studies in Hongkong were consistent with our safety and immunogenicity outcomes. Overall, our results for the first time, provided a comprehensive picture of the immunogenicity and safety of COVID-19 inacti-vated vaccines in people (especially seniors) with underlying medical conditions.

Our study has a few limitations. First, the safety data of this retrospective study was collected 14–28 days after vaccination instead of daily report. The accuracy of adverse event reports could thus be somewhat affected. Second, the immunogenicity data of this study was immune correlate of protection (CoP), which could be influenced by assessment technology platforms and might not accurately translate into clinical protection. Despite of this limitation, our results were consistent with the real-world VE against severe and mortality of COVID-19 disease for CoronaVac in Hongkong. Thus, our CoP results were useful for addressing vaccine hesitancy in people living with studied medical conditions. Third, we only considered the rough classi-fication of the chronic diseases for highly heterogeneous diseases. For example, some subtypes of cancer may have great impact on the immune system, resulting in an abnormal response to the vaccination. However, the assessment for safety and immuno-genicity of CoronaVac on HIV positive or autoimmune rheu-matic diseases (ARD) patients showed that the immunogenicity of immunosuppressed patients was significantly decreased, but at an acceptable range; and they all showed a good safety profile with the CoronaVac[17,33]. Therefore, we believe that CoronaVac will be safe in immune-related cancers but likely need further

booster doses. Fourth, the health conditions variation we assessed mainly relay on participants reported symptoms, without disease biochemical markers quantitative analysis. As we have followed up the participant's healthy condition for as long as 6 months after 2 doses of vaccine and received no report on significant alteration of healthy condition. Thus, we thought the CoronaVac vaccine has little influence on the disease progression of stable-period chronic diseases patients. In conclusion, CoronaVac vac-cination showed comparable immunogenicity and safety in individuals with and without common chronic diseases, including those ≥60 years of age. Considering that individuals with comorbidities have a significantly higher risk of progression to severe conditions or death when infected with SARS-CoV-2, we strongly recommend people with common chronic diseases get vaccinated when condition permit. Additionally, our results and Hongkong real-world investigations could provide evidence for some chronic diseases to remove from the precautions list of CoronaVac inactivated vaccine.

### Data availability
All data analyzed during this study are included in the Supplementary Data files 1–3. Supplementary Data 1 contains the source data for Fig. 2. Supplementary Data 2 contains the source data for Fig. 3. Supplementary Data 3 contains the source data for Fig. 4. Other data (such as flowcytometry data) are available from the corresponding author upon reasonable request.

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

## Acknowledgements
We thank the Yunnan Provincial Science and Technology Department (202102AA100051 and 202003AC100010, China) and Sinovac Biotech Ltd. (PRO-nCOV-4004) and Spring City Plan: the High-level Talent Promotion and Training Project of Kunming (2022SCP001) for research funding. We thank researchers and clinicians from Haikou, Wenchang and Qionghai CDC for generous support on volunteer recruitment.

## Author contributions
Z.Z., C.L., and G.Z. designed and supervised this study. Z.Z., C.L., A.L., and X.Y. wrote the manuscript. J.W., Z.F., Y.T., Y.L., T-C.Z., W.S., Y.L., M.Y., T.S., R.W., W.Q., Z.Z., F.Y., and J.H. processed the clinical samples and information. H.B., A.L., N.W., T.S., H-Y.Z., and Y-T.Z. contributed to the immunogenicity experiments. H.B. and C.L. contributed to the data analysis. T.Z. and Z.Z. contributed to adverse events evaluation.

## Competing interests
Z.Z. declares no competing non-financial interests but the following competing financial interests: this study is sponsored by Sinovac Biotech Ltd. Z.Z. on behalf of Yunnan University as an investigator has received research funding for this study from Sinovac Biotech Ltd. Gang Zeng declares competing non-financial interests as an employee of Sinovac Biotech Ltd. Other authors declare no competing interests.

## Additional information

## the Precise-CoVaccine study group

Zijie Zhang [1,4✉], Chunmei Li[1✉], Yanli Chen[1], Wei Yang[1], Xupu Ma[1], Hanfang Bi[1], Ao Li[1], Na Wan[1], Rong Wang[1,6,7], Wanting Qin[1,6,7], Xuanjing Yu[1,6,7], Zumi Zhou[1], Xinshuai Zhao[1], Xinyu Jiang[1], Wei Su[1], Tianpei Shi[1,6,7], Mei Yang[1], Qingqin Wu[1], Yating Yan[1], Lei Xing[1], Jingmei Li[1], Lipei Sun[1], Hanyi Jiao[1], Junze Wu[1], Xueyan Liu[1], Houze Yu[1], Jia Wei[4], Tai-Cheng Zhou[4], Muxian Dai[9], Fengwei Liu[9], Muhua Feng[9], Jun Hu[3], Yuemiao Zhang[1,6,10], Ying Wu[11], Dingyun You[12,13,14], Guo-Dong Wang[6,7], Zhenwang Fu[2], Guanghong Yan[12], Gangxu Xu[9], Yajing Wang[9], Lihong Zhang[15] & Liang Zhang[9]

[9]Central Lab and Liver Disease Research Center, The Affiliated Hospital of Yunnan University, 650091 Kunming, Yunnan, China. [10]Renal Division, Department of Medicine, Peking University First Hospital, Renal Pathology Center, Institute of Nephrology, Peking University, Key Laboratory of Renal Disease, Ministry of Health of China, Key Laboratory of CKD Prevention and Treatment, Ministry of Education of China; Research Units of Diagnosis and Treatment of Immune-Mediated Kidney Diseases, Chinese Academy of Medical Sciences, 100034 Beijing, China. [11]Guangdong Provincial Key Laboratory of Tropical Disease Research, Department of Biostatistics, School of Public Health, Southern Medical University, Guangdong, China. [12]School of Public Health, Kunming Medical University, Kunming, Yunnan, China. [13]Yunnan Key Laboratory of Stem Cell and Regenerative Medicine, Institute of Biomedical Engineering, Kunming Medical University, 650500 Kunming, China. [14]Yunnan Key Laboratory of Stomatology, Kunming Medical University, 650500 Kunming, Yunnan, China. [15]Department of Preventive Medicine, The Affiliated Hospital of Yunnan University, Yunnan University, 650091 Kunming, Yunnan, China.

