## [Peer Review File · Communications Medicine]

Reviewers' comments:

Reviewer #1 (Remarks to the Author):

This manuscript reports on the results of a multicenter retrospective clinical study comparing immunogenicity and reactogenicity of inactivated SARS-CoV-2 vaccine in senior population with common chronic diseases. The primary endpoint of the trial indicated safety, immunogenicity and cellular immunity memory of CoronaVac in this special population.

The manuscript is generally well written and structured, the methods used are appropriate and the conclusions drawn are reasonable. However, the authors need to address several methodological questions and queries (below).

Questions & Comments:

1. I am concerned about the major inter-group difference of systemic adverse reactions, mainly fatigue (26 [8.75%] of 297 in comorbidities, 3 [2.48%] 161 of 121 in health). Can the authors please comment detail on this occurring in 1st dose or 2nd dose and provide information on any ongoing studies to assess the safety to the vaccine.
2. Would the authors describe the diagnose of six underlying medical conditions?
3. More details are needed on the follow-up procedures. What was the basis for “chronic medical conditions” , and its development in follow-up? Were there differences in follow-up procedures for this group?
4. Were any other tests of specific T-cell response performed distinct from AIM assay? Data before vaccination are need for evaluating T cell activated target to viral antigen

Reviewer #2 (Remarks to the Author):

This manuscript reported the results of a retrospective cohort study, in which the authors evaluate the safe and immunogenicity of the inactivated covid-19 vaccine, coronaVac, in people with chronic diseases, especially the elders aged over 60 years. No vaccine-related serious adverse event was reported both for the participants with chronic diseases and the heathy control group. Statistical difference in the neutralizing antibody response between the experimental and control groups was observed for the coronary artery and chronic respiratory diseases, but such difference diminished over time. T cell immunity was detectable both in the disease and healthy groups, and there was no significant difference between them. This study has certain referring value for the vaccination of inactivated covid-19 vaccine in the persons with chronic diseases. The manuscript is well written, but there are still some major issues that should be addressed.

1. In the safety assessments, the symptoms and biochemical markers directly related to the chronic diseases were not monitored in the participants after the vaccination. It is valuable

to assess whether the vaccination affects the severity of the chronic diseases. This point should be discussed or listed as a limitation of the study.

2. Lines 203-210. Based on the data displayed in Fig. 3a, the neutralizing antibody GMTs between different age groups (40-59 vs ≥ 60 years old) were discussed. However, the data of two age groups were shown in different sub-figures in Fig. 3a. To facilitate the comparison, the data are suggested to display in the same graph.

3. Both for the safety and immunogenicity, significant differences were observed in some disease sub-groups compared to the healthy control group. However, the authors claimed "We found that CoronaVac is as safe in people with chronic diseases as that in healthy control, ...". It should be cautious in drawing such conclusions.

4. More information about the live-virus neutralization assay should be provided, such as the exact virus strain used in the assay, the provider of the virus, the experimental site, ...

5. Line 368-372. The authors stated that the neutralizing antibody in parts of the participants (95 third dose recipients) was quantified, however, the neutralization data for all participants with available serum sample were reported in the main text. Are they contradictory?

6. Are the adverse events reported in the text solicited or non-solicited? More details on the adverse event collection should be supplied in the Method section.

7. The safety data reported in the text is adverse event or adverse reaction? Adverse reaction refers to the vaccine-related adverse event.

Reviewer #3 (Remarks to the Author):

The investigators report a safety and immunogenicity study of a widely used, whole-virus, inactivated, alum-adjuvanted, 2-dose-primary-series SARS-CoV-2 vaccine, CoronaVac, developed and produced by Sinovac, in adults over 40 years of age. The vaccine has been licensed in China for over a year and a half and has been emergency-use listed by WHO for over a year. Billions of doses of the vaccine have been used around the world.

The study was an observational study in which subjects were recruited to participate 2-4 weeks after receipt of their second dose of the two-dose primary-series. Subjects were selected who had one of six comorbidities or no comorbidity (for a healthy control group). Matching between comorbidity groups and healthy subjects was on sex. Subjects were excluded if they did not complete a demographic/health survey or if they received a dose of a different brand of vaccine. Ultimately, 740 people with comorbidities and 249 healthy subjects were included. Blood was drawn upon recruitment and on day 90 and day 180 to measure neutralizing antibody levels against ancestral SARS-CoV-2 live virus and cellular immunity. Samples were considered seropositive at 1:8 or higher dilutions. Adverse events were assessed through interview (for serious AE) and through questionnaire for grades 1-3 AEs.

They found that there were no serious safety signals in the comorbidity groups or the healthy controls. They found that immunogenicity differed little between the healthy controls and the subjects with comorbidities. Neutralizing antibody levels and seropositivity decreased with time since vaccination in all groups. They concluded that "CoronaVac vaccination showed comparable effectiveness and safety in individuals with and without

common chronic diseases, including those ≥ 60 years of age.” They “strongly recommend people with common chronic diseases get vaccinated when condition permit.”

CoronaVac is an obviously important COVID-19 vaccine. There are immunogenicity and safety studies of CoronaVac in the scientific literature, but this study is of interest because it assesses immunogenicity and safety in a special population – people with comorbidities and older people. Although this is an observational study, reactogenicity was assessed retrospectively, and the endpoint is immunogenicity rather than effectiveness, the study provides very useful data that will help provide reassurance that the vaccine will work well and is safe in people with comorbidities. The conclusions are based on the data presented (with the wording exception noted below), and the recommendation follows the conclusion logically.

I have some suggestions to improve the manuscript.

The endpoint is immunogenicity – humoral and cellular. There is no agreed upon correlate of protection for COVID-19 vaccines – and a CoP would likely be different for different technology platforms and for different outcomes. Thus, it is not possible to unambiguously link this study with an assessment of effectiveness. This should be mentioned in the limitations. Additionally, there are a couple of CoronaVac vaccine effectiveness studies among people with comorbidities or older people – recent ones in Hong Kong showed that CoronaVac is highly effective in people with diabetes against severe outcomes and most effective when boosted (<https://doi.org/10.1016/j.jinf.2022.08.008>). Similar finding in people with chronic kidney disease ([doi:10.1016/j.kint.2022.07.018](https://doi.org/10.1016/j.kint.2022.07.018)), and older people ([https://doi.org/10.1016/S1473-3099\(22\)00345-0](https://doi.org/10.1016/S1473-3099(22)00345-0)). It would be useful to the readers if VE evidence that supports their findings could be discussed.

Related to the above point, the conclusion on line 315-317 should be changed from “CoronaVac vaccination showed comparable effectiveness and safety in individuals with and without common chronic diseases, including those ≥ 60 years of age” to “CoronaVac vaccination showed comparable immunogenicity and safety in individuals with and without common chronic diseases, including those ≥ 60 years of age.”

My sense is that this study shows that compared with healthy people, CoronaVac works the same in people with any of these common comorbidities and who are older and has no serious safety signal. Some of these comorbidities may be listed as precautions in the package insert in China. It would be useful for the authors to discuss in the Discussion section whether these comorbidities should be removed from lists of precautions. The providers would appreciate not having precautions listed that are not evidence based, and perhaps the evidence in this study could be used to help remove false contraindications or precautions.

The English of the manuscript is clear enough to follow, but it would benefit from some polishing to bring to journal standards. A minor point, “scleroma” would be better translated as “induration.”

Dear Editor,

We thank the reviewers for their encouraging comments and constructive suggestions. These comments/suggestions have been very helpful for improving our manuscript. We have fully revised our manuscript and listed detailed responses to reviewers as shown below. Additionally, we have added the gating strategy figure of AIM assay that was missing at the first submission in the revised supplementary file.

Reviewer(s)' Comments to Author:

Reviewer #1 :

This manuscript reports on the results of a multicenter retrospective clinical study comparing immunogenicity and reactogenicity of inactivated SARS-CoV-2 vaccine in senior population with common chronic diseases. The primary endpoint of the trial indicated safety, immunogenicity and cellular immunity memory of CoronaVac in this special population.

The manuscript is generally well written and structured, the methods used are appropriate and the conclusions drawn are reasonable. However, the authors need to address several methodological questions and queries (below).

Questions & Comments:

1. I am concerned about the major inter-group difference of systemic adverse reactions, mainly fatigue (26 [8.75%] of 297 in comorbidities, 3 [2.48%] 161 of 121 in health). Can the authors please comment detail on this occurring in 1st dose or 2nd dose and provide information on any ongoing studies to assess the safety to the vaccine.

Response: We thank the reviewer for these two important questions. The major inter-group difference of systemic adverse events was fatigue in adults (40-59 years old). The incidence of fatigue in adults (40-59 years old) group was 20 (6.73%) of 297 in the comorbidities group after the first dose, with significant difference from the health control (1 [0.83%] of 121, $P=0.0115$). There was no significant difference between comorbidities and health groups after the second dose with 12 (4.04%) of 297 in the comorbidities group and 2 [1.65%] of 121 in the health control ($P=0.3678$). We have added this results in line 161-166 of the revised manuscript file.

Regarding the safety assessment, the newly published real-world study of CoronaVac inactivated vaccine in Hongkong can provide valuable info. Cheng FWT. et al shown there was no increased risk of overall adverse event for CoronaVac among individuals with chronic kidney diseases, diabetes and seniors (≥ 65 years old) (<https://doi.org/10.1016/j.kint.2022.07.018>). Kang W. et al reported the incidence of adverse events was low for CoronaVac in adults with active cancer or a history of cancer (<https://doi.org/10.1186/s13045-022-01265-9>). These results are consistent with ours.

2. Would the authors describe the diagnose of six underlying medical conditions?

Response: We thank the reviewer for this great suggestion. The diagnose of six underlying medical conditions in our study were not described thoroughly enough, and we have added it to the Methods section in the revised manuscript and the protocol file (5.1 study design) as shown below:

Methods: “We recruited participants who have received 2 doses of CoronaVac inactivated vaccine with 3-5 weeks of dose interval and were at the 14th-28th day after the second dose at the time of enrollment. Participants were eligible if they were 40 years of age or older, healthy, or diagnosed with any of the 6 most common chronic diseases: hypertension, diabetes mellitus (DM), coronary artery disease (CAD), chronic respiratory disease (CRD), obesity, and cancer, and were able to understand and complete the questionnaires. The diagnose of six underlying medical conditions in our study were mainly based on the diagnosis of local hospital according to National clinical practice guidelines in China. Specifically, we screened potential eligible participants through the CDC database of chronic diseases and vaccination. We then recruit the volunteers who registered as healthy or with at least one of the diseases we studied by telephone interview. During the telephone visit, we further narrowed down to people who have a proof of chronic diseases diagnosis of interest from a local hospital. At the first visit, we checked their case history and asked about their treatment and disease status. People were considered as having underlying medical conditions if they were once diagnosed as one of the 5 common chronic diseases (hypertension, DM, CAD, CRD and cancer), and the disease was not in the acute phase during the recruitment, and maintained the regular treatment during vaccination. The obesity was diagnosed as $BMI \geq 28.0 \text{ kg/m}^2$. Furthermore, they were excluded according to established criteria: had been infected by SARS-CoV-2; had received non-CoronaVac vaccine; with severe mental and neurological diseases; with any other factors unsuitable for clinical observation.”.

5.1 study design: “We will recruit participants who have received 2 doses of CoronaVac with 3-5 weeks of dose interval and are at the 14th-28th day after the second dose at the time of enrollment. Participants are eligible if they are 40 years of age or older, healthy, or diagnosed with any of the 6 most common chronic diseases: hypertension, diabetes mellitus (DM), coronary artery disease (CAD), chronic respiratory disease (CRD), obesity, and cancer, and can understand and complete the questionnaires. The diagnose of six underlying medical conditions in our study will mainly based on the diagnosis of local hospital according to National clinical practice guidelines in China. Specifically, we will screen potential eligible participants through the CDC database of chronic diseases and vaccination. We then will recruit the volunteers who registered as healthy or with at least one of the diseases we studied by telephone interview. During the telephone visit, we will further narrow down to people who have a proof of chronic diseases diagnosis of interest from a local hospital. At the first visit, we will check their case history and asked about their treatment and disease status. People will considered as having underlying medical conditions if they were once diagnosed as one of the 5 common chronic diseases

(hypertension, DM, CAD, CRD and cancer), and the disease is not in the acute phase during the recruitment, and will maintain the regular treatment during vaccination. The obesity was diagnosed as $BMI \geq 28.0 \text{ kg/m}^2$. They will be excluded according to established criteria: had been infected by SARS-CoV-2; had received non-CoronaVac vaccine; with severe mental and neurological diseases; with any other factors unsuitable for clinical observation.”.

3. More details are needed on the follow-up procedures. What was the basis for “chronic medical conditions”, and its development in follow-up? Were there differences in follow-up procedures for this group?

Response: We thank the reviewer for this great suggestion and questions. Our follow-up procedures was not described thoroughly enough. Indeed, we have recorded the changes in their chronic medical condition for the diseases group at the first visit and follow-up procedures. We have revised the description of the first visit and follow-up in the Protocol file as shown below:

“5.4 First visit

The first visit will be conducted at 14-28 days post the second-dose vaccination. We will check their case history and asked about their treatment and disease status. People will considered as having underlying medical conditions if they were once diagnosed as one of the 5 common chronic diseases (hypertension, DM, CAD, CRD and cancer), and the disease is not in the acute phase during the recruitment, and will maintain the regular treatment during vaccination. The obesity will diagnosed as $BMI \geq 28.0 \text{ kg/m}^2$. For eligible participants, the blood samples will be collected to explore the GMT of neutralizing antibodies and cellular responses. We will record all adverse events, including serious/severe adverse events, and the changes in their chronic medical condition, occurred within 14 days after each dose.

5.5 Follow-up

Eligible participants will be followed up at 3 and 6 months after the second inactivated vaccine dose. Blood samples will be collected at the time of 3 and 6 months after the second dose of vaccine to explore the GMT of neutralizing antibodies and cellular responses. Non-routine health-care contacts including hospitalization and the reasons for these healthcare visits will be recorded. Additionally, disease condition variation will be recorded for participants underlying medical conditions.

Type of contact	Visit 1 (Blood collection)	Visit 2 (Blood collection)	Visit 3 (Blood collection)	Passive follow-up
Time points	Day 14-28 post the second dose vaccination	3 months post vaccination	6 months post vaccination	Up to 6 months post vaccination
Eligibility review	√			

Review of signed consent form	✓			
Blood draw	✓	✓	✓	
Participant reporting of adverse events or serious adverse events	✓			✓
Participant reporting of their chronic medical conditions variation	✓			✓

”

4. Were any other tests of specific T-cell response performed distinct from AIM assay? Data before vaccination are need for evaluating T cell activated target to viral antigen.

Response: We thank the reviewer for this interesting question. In this study, we mainly used the AIM assay in this study to profile SARS-CoV-2-specific CD4⁺ and CD8⁺ T cells separately.

We total agree with the reviewer that data before vaccination are of great interest! Because this trial is a retrospective study focusing on comparing immunogenicity of CoronaVac in healthy and diseased people, we didn't collect serum sample before vaccination. However, in another unpublished study where we focused on the pre-existing immunity against SARS-CoV-2 before vaccination/infection, we performed an interferon- γ (IFN γ) and interleukin-4 (IL-4) and Granzyme B FluoroSpot assay to evaluate T cell responses using the PBMCs collected before vaccination and following the second dose with overlapping peptide pools to the wild-type SARS-CoV-2 protein as stimuli. The number of peptide-specific T cells was calculated by subtracting the unstimulated control. The frequency of having peptide-specific T cells (by counting IFN γ -secreting spots) in unvaccinated individuals varies from 25% (16 of 64) to 48% (31 of 64) depending on the strengency parameter of FluoroSpot data analysis. We also found the number of IFN γ / IL-4 dual-secreting spots in unvaccinated individuals correlates with the number of IFN γ / Granzyme B dual-secreting spots (R=0.7, p=0.037) and IL-4 secreting spots (R=0.69, p=0.039) after the second dose.

Reviewer #2:

This manuscript reported the results of a retrospective cohort study, in which the authors evaluate the safe and immunogenicity of the inactivated covid-19 vaccine, coronaVac, in people with chronic diseases, especially the elders aged over 60 years. No vaccine-related serious adverse event was reported both for the participants with chronic diseases and the healthy control group. Statistical difference in the neutralizing antibody response between the experimental and control groups was observed for the coronary artery and chronic respiratory diseases, but such difference diminished over time. T cell immunity was detectable both in the disease and healthy groups, and there was no significant difference between them. This study has certain referring value for the vaccination of inactivated covid-19 vaccine in the persons with chronic diseases. The manuscript is well written, but there are still some major issues that should be addressed.

1. In the safety assessments, the symptoms and biochemical markers directly related to the chronic diseases were not monitored in the participants after the vaccination. It is valuable to assess whether the vaccination affects the severity of the chronic diseases. This point should be discussed or listed as a limitation of the study.

Response: We thank the reviewer for the great suggestion. Indeed, we have followed up the participant's healthy condition for as long as 6 months after 2 doses of vaccine and received no report on significant alteration of healthy condition. Thus, our data suggested that CoronaVac inactivated vaccine showed little influence on the stable-period chronic diseases. We have added the discussion about the limitation of the safety assessment measurements without monitored biochemical markers in the revised manuscript file as shown below:

“Fourth, the health conditions variation we assessed mainly relay on participants reported symptoms, without disease biochemical markers quantitative analysis. As we have followed up the participant's healthy condition for as long as 6 months after 2 doses of vaccine and received no report on significant alteration of healthy condition. Thus, we thought the CoronaVac vaccine has little influence on the disease progression of stable-period chronic diseases patients.”.

2. Lines 203-210. Based on the data displayed in Fig. 3a, the neutralizing antibody GMTs between different age groups (40-59 vs ≥ 60 years old) were discussed. However, the data of two age groups were shown in different sub-figures in Fig. 3a. To facilitate the comparison, the data are suggested to display in the same graph.

Response: We thank the reviewer for this good advice. We have revised the Fig. 3a as suggested by the reviewer in the revised manuscript, and show the updated figure below:

3. Both for the safety and immunogenicity, significant differences were observed in some disease sub-groups compared to the healthy control group. However, the authors claimed “We found that CoronaVac is as safe in people with chronic diseases as that in healthy control, ...”. It should be cautious in drawing such conclusions.

Response: We gratefully appreciate this comment. Although a few disease sub-groups showed significant differences with healthy control: ① senior CRD group showed significantly higher incidence of injection-site pain ($P=0.0404$); ② adult hypertension group ($P=0.0124$), DM group ($P=0.0403$) and cancer group ($P=0.0152$) showed significantly higher incidences of fatigue; ③adult CRD group ($P=0.0405$) showed significant higher incidence of headache; these adverse events were mostly mild (grade 1) in severity and recovered within 48 hours without intervention. Thus, these adverse events were of little safety concern of vaccine in the people with chronic diseases. Per suggestion by the reviewer, to be more precise, we have corrected this statement in the revised manuscript as shown below:

“ We found that CoronaVac is **almost** as safe in people with chronic diseases as that in

healthy control.”.

4. More information about the live-virus neutralization assay should be provided, such as the exact virus strain used in the assay, the provider of the virus, the experimental site, ...

Response: We thank the reviewer for this suggestion. More details of the live-virus neutralization assay were specified in the Methods section of revised manuscript as follows:

“The neutralizing antibodies to live SARS-CoV-2 were quantified at the central lab and liver disease research center of the affiliated hospital of Yunnan university, using the gold standard of antibody titration: 50 µL serum was first inactivated at 56°C for 30 minutes and serially diluted with cell culture medium in two-fold steps. The diluted serum was then incubated with equal volume of live wild type SARS-CoV-2 virus (virus titers: 6.0 IgCCID 50/mL; passage: P5; GenBank number: MT407649.1) for 2 hours at 36.5°C. Vero cells were then added to the serum-virus mix and incubated at 36.5°C for 5 days. Neutralizing antibody titer was calculated by the highest dilution number of 50% protective condition.”.

5. Line 368-372. The authors stated that the neutralizing antibody in parts of the participants (95 third dose recipients) was quantified, however, the neutralization data for all participants with available serum sample were reported in the main text. Are they contradictory?

Response: We are really sorry for our careless mistakes and thanks for reviewer’s reminder. We have deleted the whole sentence "microcytopathogenic effect assay with live virus to quantify the neutralizing antibody in 95 third dose recipients (subjects of live virus test were selected according to the availability of serum sample). " in the revised manuscript file.

6. Are the adverse events reported in the text solicited or non-solicited? More details on the adverse event collection should be supplied in the Method section.

Response: We thank the reviewer for this good advice. We have added the solicited or non-solicited adverse events description in the Method section of the revised manuscript file as shown below:

“Adverse events collection

Participants who came for serum tests 14-28 days after two-dose vaccination were assessed for adverse events within vaccination period. Adverse events including solicited and non-solicited. Solicited injection-site adverse events containing pain, induration, swelling, redness, rash and itching; Solicited systemic adverse events including acute allergy reaction, skin & mucosa abnormalities, diarrhoea, anorexia, vomiting, nausea, muscle pain, headache, cough, fatigue and fever. Participant were first screened for grade 3 (daily activity significantly affected, medical attention required, and hospitalization may be necessary) or grade 4 (potential life threat, daily activity severely limited and hospitalization required) solicited and non-solicited adverse events by researcher. Participants were then requested to fill a survey to report grade 1 (short or mild reactions without interfering daily activity) and grade 2 (daily activity mildly interfered, simple

treatment or no treatment was necessary) solicited and non-solicited adverse events.”.

7. The safety data reported in the text is adverse event or adverse reaction? Adverse reaction refers to the vaccine-related adverse event.

Response: We thank the reviewer for this great suggestion. We fully agree that most of the site effects we reported is adverse events. We have replaced all "adverse reaction" or "adverse reaction event" which is not directly relate to vaccine to "adverse event" in the revised manuscript and the supplementary file.

Reviewer #3:

The investigators report a safety and immunogenicity study of a widely used, whole-virus, inactivated, alum-adjuvanted, 2-dose-primary-series SARS-CoV-2 vaccine, CoronaVac, developed and produced by Sinovac, in adults over 40 years of age. The vaccine has been licensed in China for over a year and a half and has been emergency-use listed by WHO for over a year. Billions of doses of the vaccine have been used around the world.

The study was an observational study in which subjects were recruited to participate 2-4 weeks after receipt of their second dose of the two-dose primary-series. Subjects were selected who had one of six comorbidities or no comorbidity (for a healthy control group). Matching between comorbidity groups and healthy subjects was on sex. Subjects were excluded if they did not complete a demographic/health survey or if they received a dose of a different brand of vaccine. Ultimately, 740 people with comorbidities and 249 healthy subjects were included. Blood was drawn upon recruitment and on day 90 and day 180 to measure neutralizing antibody levels against ancestral SARS-CoV-2 live virus and cellular immunity. Samples were considered seropositive at 1:8 or higher dilutions. Adverse events were assessed through interview (for serious AE) and through questionnaire for grades 1-3 AEs.

They found that there were no serious safety signals in the comorbidity groups or the healthy controls. They found that immunogenicity differed little between the healthy controls and the subjects with comorbidities. Neutralizing antibody levels and seropositivity decreased with time since vaccination in all groups. They concluded that “CoronaVac vaccination showed comparable effectiveness and safety in individuals with and without common chronic diseases, including those ≥ 60 years of age.” They “strongly recommend people with common chronic diseases get vaccinated when condition permit.”

CoronaVac is an obviously important COVID-19 vaccine. There are immunogenicity and safety studies of CoronaVac in the scientific literature, but this study is of interest because it assesses immunogenicity and safety in a special population – people with comorbidities and older people. Although this is an observational study, reactogenicity was assessed retrospectively, and the endpoint is immunogenicity rather than effectiveness, the study provides very useful data that will help provide reassurance that the vaccine will work well and is safe in people with comorbidities. The conclusions are based on the data presented (with the wording exception noted below), and the recommendation follows the conclusion logically.

I have some suggestions to improve the manuscript.

The endpoint is immunogenicity – humoral and cellular. There is no agreed upon correlate of protection for COVID-19 vaccines – and a CoP would likely be different for different technology platforms and for different outcomes. Thus, it is not possible to unambiguously link this study with an assessment of effectiveness. This should be mentioned in the limitations. Additionally, there are a couple of CoronaVac vaccine effectiveness studies

among people with comorbidities or older people – recent ones in Hong Kong showed that CoronaVac is highly effective in people with diabetes against severe outcomes and most effective when boosted (<https://doi.org/10.1016/j.jinf.2022.08.008>). Similar finding in people with chronic kidney disease ([doi:10.1016/j.kint.2022.07.018](https://doi.org/10.1016/j.kint.2022.07.018)), and older people ([https://doi.org/10.1016/S1473-3099\(22\)00345-0](https://doi.org/10.1016/S1473-3099(22)00345-0)). It would be useful to the readers if VE evidence that supports their findings could be discussed.

Related to the above point, the conclusion on line 315-317 should be changed from “CoronaVac vaccination showed comparable effectiveness and safety in individuals with and without common chronic diseases, including those ≥ 60 years of age” to “CoronaVac vaccination showed comparable immunogenicity and safety in individuals with and without common chronic diseases, including those ≥ 60 years of age.”

My sense is that this study shows that compared with healthy people, CoronaVac works the same in people with any of these common comorbidities and who are older and has no serious safety signal. Some of these comorbidities may be listed as precautions in the package insert in China. It would be useful for the authors to discuss in the Discussion section whether these comorbidities should be removed from lists of precautions. The providers would appreciate not having precautions listed that are not evidence based, and perhaps the evidence in this study could be used to help remove false contraindications or precautions.

The English of the manuscript is clear enough to follow, but it would benefit from some polishing to bring to journal standards. A minor point, “**scleroma**” would be better translated as “**induration**.”

Response: We gratefully appreciate the reviewer for these valuable suggestions. We have made correction according to the Reviewer’s suggestions in the revised manuscript file as shown below:

(1) We thank reviewer for reminding those real-world vaccine effectiveness (VE) studies, which are very valuable reflections for our study. We have added those real-world evidences to the Discussion section as follow:

“Our analyses for CoronaVac, the most widely administered inactivated SARS-CoV-2 vaccine, demonstrated consistent trend with those analyzed in mRNA vaccine trials. Furthermore, the real-world study in Hongkong showed two doses of CoronaVac vaccine had 69.9% [64.4-74.6] severe disease and death protection effectiveness among adults aged 60 years or older, and three doses of CoronaVac had provided high levels of protection against severe or fatal outcomes (97.9% [97.3-98.4]).²⁹ For people with underlying medical conditions, the real-world studies for patient with DM and chronic kidney diseases all showed no increased risk of adverse events, and showed high vaccine effectiveness (VE) against severe and mortality of COVID-19 disease post two-dose CoronaVac vaccine.^{30,31} The results of these real-world studies in Hongkong were consistent with our safety and immunogenicity outcomes. Overall, Our results for the first

time, provided a comprehensive picture of the immunogenicity and safety of COVID-19 inactivated vaccines in people (especially seniors) with underlying medical conditions.”.

(2) We fully agree with the reviewer about the limitation of immunogenicity as the surrogate endpoint to assess the vaccine efficacy. We have added this limitation in the Discussion section of the revised manuscript file as follow:

“Second, the immunogenicity data of this study was immune correlate of protection (CoP), which could be influenced by assessment technology platforms and might not accurately translate into clinical protection. Despite of this limitation, our results were consistent with the real-world VE against severe and mortality of COVID-19 disease for CoronaVac in Hongkong. Thus, our CoP results were useful for addressing vaccine hesitancy in people living with studied medical conditions.”.

(3) We have replaced the sentence “CoronaVac vaccination showed comparable effectiveness and safety in individuals with and without common chronic diseases, including those ≥ 60 years of age” as suggested to:

“CoronaVac vaccination showed comparable immunogenicity and safety in individuals with and without common chronic diseases, including those ≥ 60 years of age.”.

(4) We gratefully appreciate for the reviewer's suggestion. As the CoronaVac inactivated vaccine showed good safety and effectiveness (immunogenicity) in people with chronic diseases (including those ≥ 60 years of age) in our study and the Hongkong real-world investigations, we fully agree that some of these comorbidities could be removed from the precautions list, based on current available data. We have added this suggestion in the Discussion section of the revised manuscript file as follow:

“Considering that individuals with comorbidities have a significantly higher risk of progression to severe conditions or death when infected with SARS-CoV-2, we strongly recommend people with common chronic diseases get vaccinated when condition permit. Additionally, our results and Hongkong real-world data provided evidence to support removal of some chronic diseases from the precautions list of CoronaVac inactivated vaccine.”.

(5) We appreciate the reviewer's language polishing suggestion. The manuscript have been double-checked, and the typos and grammar errors we found have been corrected. The word “scleroma” all replaced by “induration” in the fig.2 and the tables of the revised manuscript and the supplementary file.

REVIEWERS' COMMENTS:

Reviewer #1 (Remarks to the Author):

The authors did lots of works in this revision. I think it is satisfied for publication.

Reviewer #2 (Remarks to the Author):

My concerns have been addressed in the revised manuscript. The manuscript could be accepted for publication.

Reviewer #3 (Remarks to the Author):

I was one of the original reviewers of the manuscript. The authors have adequately addressed my comments and suggestions. I have no new suggestions or comments except for the minor suggestion, below.

In the abstract, the authors report a finding that “We found that CoronaVac is almost as safe in people with chronic diseases as that in healthy control, without serious adverse event reported in this study.” The word “almost” was added in the revised version. I recommend removing the word “almost” because the basis for “almost” is not a safety signal, it is a reactogenicity signal (fatigue following dose 1). Their study found no safety signal, and therefore the phrase “almost as safe” could serve as a false deterrent to vaccination. The phrase makes it sound like the vaccine is not as safe in people with comorbidities, but that is not what their study found. Instead, if they want to highlight this difference in reactogenicity, they could consider saying, “We found that CoronaVac was associated with a higher reported rate of fatigue following dose 1 in people with chronic diseases than in people without chronic diseases, but there were no serious adverse events observed in any study group.”

Dear reviewer,

We thank you for the affirmation and help with our manuscript, and thanks reviewers 3 for the constructive suggestions. These suggestions have been very helpful in improving our manuscript. We have fully revised our manuscript and listed detailed responses to reviewers as shown below.

REVIEWERS' COMMENTS:

Reviewer #1 (Remarks to the Author):

The authors did lots of works in this revision. I think it is satisfied for publication.

Reviewer #2 (Remarks to the Author):

My concerns have been addressed in the revised manuscript. The manuscript could be accepted for publication.

Reviewer #3 (Remarks to the Author):

I was one of the original reviewers of the manuscript. The authors have adequately addressed my comments and suggestions. I have no new suggestions or comments except for the minor suggestion, below.

In the abstract, the authors report a finding that “We found that CoronaVac is almost as safe in people with chronic diseases as that in healthy control, without serious adverse event reported in this study.” The word “almost” was added in the revised version. I recommend removing the word “almost” because the basis for “almost” is not a safety signal, it is a reactogenicity signal (fatigue following dose 1). Their study found no safety signal, and therefore the phrase “almost as safe” could serve as a false deterrent to vaccination. The phrase makes it sound like the vaccine is not as safe in people with comorbidities, but that is not what their study found. Instead, if they want to highlight this difference in reactogenicity, they could consider saying, “We found that CoronaVac was associated with a higher reported rate of fatigue following dose 1 in people with chronic diseases than in people without chronic diseases, but there were no serious adverse events observed in any study group.”

Response: The addition of “almost” in our revised abstract is indeed ill-considered. We gratefully appreciate the reviewer pointing it out. We had rewritten the abstract section into two parts: the abstract section and plain language summary section, upon request by “Final Revision Instructions”, in our revised manuscript. For the abstract section, we replaced the “almost” statement with details about the safety results (highlighted in bold font). For the plain language summary section, we just removed the word “almost” due to the word limitation and considering the accessibility to members of the general public (highlighted by bold font). All changes are highlighted in blue font as shown below:

“Abstract

Background

People living with chronic disease, particularly seniors (≥ 60 years old), made up of most severe symptom and death cases among SARS-CoV-2 infected patients. However, they are lagging behind in the national COVID-19 vaccination campaign in China due to the uncertainty of vaccine safety and effectiveness. Safety and immunogenicity data of COVID-19 vaccines in people with underlying medical conditions are needed to address the vaccine hesitation in this population.

Methods

We included participants (≥ 40 years old) received two doses of CoronaVac inactivated vaccines (3-5 weeks interval), been healthy or with at least one of 6 common chronic diseases. The incidence of adverse events after vaccination was monitored as the safety data. The vaccine immunogenicity was studied by neutralizing antibodies and SARS-CoV-2-specific T cell responses post vaccination.

Results

Here we show that **chronic diseases are associated with a higher rate but mild fatigue following the first dose of CoronaVac**. By day 14-28 post vaccination, the neutralizing antibody level show no significant difference between disease groups and healthy control, except for the coronary artery disease ($p=0.0287$) and chronic respiratory disease group ($p=0.0416$) showing moderate reductions. Such differences diminish by day 90 and 180. Most people show detectable SARS-CoV-2-specific T cell response at day 90 and day 180 without significant difference between disease groups and healthy control.

Conclusions

Our results highlight the comparable safety, immunogenicity and cellular immunity memory of CoronaVac in seniors and people living with chronic diseases, addressing vaccine hesitancy for this special population.

Plain language summary

People living with chronic diseases, particularly seniors (≥ 60 years old), made up of most severe symptom and death cases among SARS-CoV-2 infected patients. However, they are lagging behind in the national COVID-19 vaccination campaign in China due to the uncertainty of vaccine safety and effectiveness. Here we show that **the inactivated COVID-19 vaccine, CoronaVac, is as safe in this special population as that in healthy counterpart**. Although the immunogenicity is slightly different in subgroup of some diseases compared with that of the healthy population, the overall trend is consistent. Overall, our results highlight the comparable safety, immunogenicity and cellular immune memory of CoronaVac in people living with chronic diseases.”